# Biomechanical Analysis of Femoral Stem Features in Hinged Revision TKA with Valgus or Varus Deformity: A Comparative Finite Elements Study

**Edoardo Bori** and **Bernardo Innocenti** *

BEAMS Department (Bio Electro and Mechanical Systems), Université Libre de Bruxelles, 1050 Brussels, Belgium
* Correspondence: bernardo.innocenti@ulb.be; Tel.: +32-(0)-2650-3531

**Featured Application: The present study intends to help and support clinicians and specialists in the decision process concerning the length and type of stems to use in case varus–valgus deformity is present in a patient.**

**Abstract:** Hinged total knee arthroplasty (TKA) is a valid option to treat patients during revision of an implant; however, in case of varus/valgus deformity, the force transmission from the femur to the tibia could be altered and therefore the performance of the implant could be detrimental. To be able to evaluate this, the goal of this study was to investigate, using a validated finite element analysis, the effect of varus/valgus load configurations in the bones when a hinged TKA is used. In detail, short and long stem lengths (50 mm, and 120 mm), were analyzed both under cemented or press-fit fixation under the following varus and valgus deformity: 5°, 10°, 20°, and 30°. The main outputs of the study were average bone stress in different regions of interest, together with tibio-femoral contact pressure and force. Results demonstrated that changes in the varus or valgus deformity degrees induce a change in the medio-lateral stress and force distribution, together with a change in the contact area. The effect of stem length and cement do not alter the tibio-femoral contact biomechanics but its effect is mainly localized in the distal femoral region, and it is negligible in the proximal regions.

**Keywords:** hinged TKA; varus deformity; valgus deformity; finite element; stem length; cemented; press fit; biomechanics





## 1. Introduction

Achieving the correct alignment in total knee arthroplasty (TKA) is an important factor to restore biomechanical functions of the knee (such as soft tissue balancing and joint line) and prevent early implant failure, thus guaranteeing long-term survival of the implant [1–3]. In detail, the alignment of the prosthesis components on the frontal plane strongly influences the distribution of tibio-femoral contact forces between the two tibial plateau compartments: biomechanical studies report that varus misalignment causes overload on the medial side while valgus misalignment leads to increased tibio-femoral force in the lateral side [4]. As a result of these deviations, the non-physiological loading of the knee enhances degeneration of the joint and causes overloads on the bone-implant interface and in the bone itself.

Alterations of the natural alignment, therefore, may lead to early aseptic loosening, polyethylene wear, erroneous patellofemoral tracking, instability and infection [2,5]; all of these factors are usually associated with TKA failure, which consequently leads to the need of a revision total knee replacement.

Together with alignment, the choice of an adequate level of constraint is essential to ensure successful clinical outcomes and long-term TKA survival [6]; this is even more important when dealing with revision surgeries.

The rotating hinge (RH) prosthesis represents a common solution among the different models available for revision procedures, as thanks to this degree of freedom around the tibial axis, this model is able to achieve reduced shear stress at the bone-cement interface, compared to other devices [7–11]. It is important to mention that this kind of implant can also be used as an alternative to primary TKA, in case of complex situations, such as extreme joint deformities [7,12].

The main issues addressed with the use of a RH TKA are therefore large ligamentous instability, severe bone loss, distal femoral or proximal tibial defects (resulting from injury or tumor) and severe varus or valgus deformities [13–15].

The design of a rotating hinge prosthesis usually consists of a femoral component and a tibial one (constrained among each other with a rotating hinge mechanism) and of a polyethylene insert placed between these two elements, in order to prevent luxation without reducing the range of motion [16].

The current designs are available in multiple sizes and present modularity, with sets of different features/stem lengths to allow for the different possible fixation techniques; this variety of possibilities represents thus a worthwhile opportunity, especially in revision patients where deficiencies of the bone cannot be totally predicted [17].

Stem length, among the other parameters, covers a crucial role in the overall success of the reconstruction. Usually, the choice of which length to use is largely determined by the stem fixation technique selected: in modern revision TKA, the options are usually fully cemented and press-fit fixation [18–21].

Despite its advantages, this kind of implant also implies the relative risks of mechanical failure and infection [9,22–24], which can furthermore increase in case of eventual patient joint deformities. The level of constraint provided by the rotating hinge, paired with the use of different stems and/or fixation approached, may therefore lead to eventual over-constraining issues affecting the tibio-femoral and/or the bone-implant interactions; a more comprehensive guideline on the suggested use and optimal configuration of these implants (when mainly aimed to address high deformities) is however still missing.

The present study intends to help and support clinicians and specialists, assisting the decision process concerning the length and type of stems to use in case varus–valgus deformity is present in a patient. A numerical biomechanical study was therefore performed in order to evaluate the effects and performances of different configurations of RH TKA prosthesis in patients with severe valgus and varus deformities, in terms of distribution of forces and relative contact areas on the polyethylene insert when different lengths of femoral stems are used, considering both fixation types (cemented and press-fit) and in terms of average bone stress in different regions of interests.

## 2. Materials and Methods

The model was developed based on an already validated and published knee finite element model [25–27].

Finite elements analysis approach was selected since it guarantees a great comparative potential, allowing to change any single parameter of the model in order to precisely analyze its influence on the outcomes while leaving the other boundary conditions unchanged [28–30].

The following features were implemented in the models.

### 2.1. Geometry & Configurations

The three-dimensional model of a femur, divided in cortical and cancellous bone, was obtained from CT scans of a right-side Sawbone synthetic bone (Sawbones Europe AB, Limhamn, Sweden), following an approach widely used for numerical and experimental tests [2,31–36].

A right side, medium size endo-model rotating hinge knee prosthesis (Waldemar Link GMBH & Co., Hamburg, Germany) was considered for the study and the relative geometries were obtained from a previous study [21].

Different angles of valgus and varus deformities of the femur were considered (5°, 10°, 20°, and 30°, considering a 0° configuration as the physiological control) and the relative models were implemented. For each configuration, two femoral stem lengths (50, 120 mm) and two types of fixations (cemented or press-fit) were tested; tibial tray component and polyethylene insert sizes were kept the same in each configuration, together with the tibial stem of 50 mm length. For each of the configurations analyzed, the prosthesis was virtually implanted into the femoral bone following the manufacturer's surgical technique (therefore considered to be the ideal positioning of the implant, according to the patient configuration [21]). In the cemented configuration, the stem was positioned in the center of the intramedullary canal and surrounded by a homogeneous cement mantle, obtained by filling a previously reamed hole and subtracting the stem volume, simulating thus the ideal cementing technique [32].

### 2.2. Material Properties

According to the literature [2,32,37–39], the materials used in this study were assumed to be linear elastic; this was chosen to obtain a better approximation of all materials, in order to achieve a qualitative comparison among different configurations [2]. In particular, the material for the femoral component and the tibial tray was considered as cobalt-chromium (CoCr), whereas the material of the tibial insert was ultra-high-molecular-weight-polyethylene (UHMWPE). These materials were assumed to be homogeneous and isotropic [2,32,40]. The material used for the bone cement (Polymethyl-methacrylate, PMMA) in the relative configurations was considered homogeneous and isotropic [2,11]. The material properties in terms of Young's modulus (E) and Poisson's ratio (ν) are available in Table 1 [2,32,37].

**Table 1.** Material properties of the implant components: CoCr = cobalt-chromium alloy, UHMWE = ultra-high-molecular-weight-polyethylene, PMMA = polymethyl-methacrylate.

| Material | Material Model | Elastic Modulus (MPa) | Poisson Ratio |
| --- | --- | --- | --- |
| CoCr | Isotropic | 240,000 | 0.30 |
| UHMW | Isotropic | 685 | 0.40 |
| PMMA | Isotropic | 3000 | 0.30 |

The cortical bone, according to previous studies, was considered transversely isotropic [2,8,41,42]; the cancellous bone was instead considered linear isotropic [2,32,43]. The material properties used are reported in Table 2.

**Table 2.** Material properties of the femoral bone used for all models; the third axis was taken parallel with the anatomical axis of the femur.

| Material | Material Model | Elastic Modulus (MPa) | | | Poisson Ratio | | |
| --- | --- | --- | --- | --- | --- | --- | --- |
| | | $E_1$ | $E_2$ | $E_3$ | $\nu_{12}$ | $\nu_{23}$ | $\nu_{31}$ |
| Cortical Bone | Transversely Isotropic | 11,500 | 11,500 | 17,000 | 0.51 | 0.31 | 031 |
| Cancellous Bone | Isotropic | 2130 | | | 0.30 | | |

### 2.3. Finite Element Analysis and Boundary Conditions

Abaqus/Standard version 6.19 (Dassault Systèmes, Vélizy-Villacoublay, France) was used to assemble all parts of the prosthesis with femur bone, in order to perform all the finite element simulations.

Each model was meshed using tetrahedral elements with a size between 1 mm and 3 mm. A refinement of mesh was performed in the contact area of the internal components of the implant, and in the tibio-femoral and bone-implant interface sections to make sure that the selected mesh was the proper one to achieve the sought after results: to check the

quality of the mesh and the proper size, a convergence test was thus performed [2,20] and is available in the Supplementary Materials.

According to the previously published study [21], surface-to-surface contacts were implemented for the definition of all contacts, i.e., among the components involved in the hinge mechanism, between the insert and the femoral component of the prosthesis and between the implant and the bone.

Each configuration analyzed was tested under the same total load conditions, i.e., the proximal part of the femur constrained and a total vertical compressive force of 1000 N applied to the inferior face of the tibial tray. The subdivision of this total force on the two sides of the tibial tray was then used as a parameter in order to model the different levels of varus/valgus deformities: a simplified model in the literature [11] allowed indeed to obtain the percentage of force distribution in the medial and lateral regions of the tibial component for each angle of deformity (based on the equilibrium of forces and moments in the frontal plane), and was therefore taken as reference. The percentages of force distribution in the medial and lateral compartments obtained from this mathematical model are reported in Table 3.

**Table 3.** Percentages of the total force applied in the different configurations analyzed in the study [11].

| Configuration | Medial Force | Lateral Force |
|---|---|---|
| 30° Varus | 91% | 9% |
| 20° Varus | 82% | 18% |
| 10° Varus | 64% | 36% |
| 5° Varus | 60% | 40% |
| Well Aligned | 55% | 45% |
| 5° Valgus | 43% | 57% |
| 10° Valgus | 38% | 62% |
| 20° Valgus | 8% | 92% |
| 30° Valgus | 1% | 99% |

These percentages were therefore used in the present study to determine the subdivision of the total force on the medial and lateral sides, and the resulting forces were applied on the respective side of the tibial tray plateau (see Figure 1), as input loads for the FE simulations.

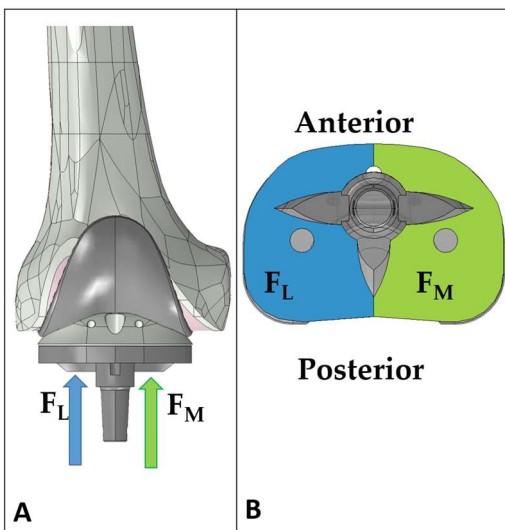

**Figure 1.** Load conditions applied to the model, (**A**) frontal view, reporting the direction of the forces applied; (**B**) distal view, reporting the area used for the applied force. FL = lateral force (in blue), FM = medial force (in green).

Each varus/valgus aligned configuration was statically analyzed in full extension, analyzing two femoral stem lengths (50, and 120 mm) and considering both types of fixations examined (cemented or press-fit).

For all of the configurations tested, the output for the finite element analysis was the medial and lateral tibio-femoral contact force and the relative contact area in the polyethylene insert, together with the average von Mises stress [44–48] in different regions of interest of the bone; in detail, in agreement with previous studies [20,21], the femoral shaft was subdivided in eight regions of interest, each with a height of 30 mm (Figure 2).

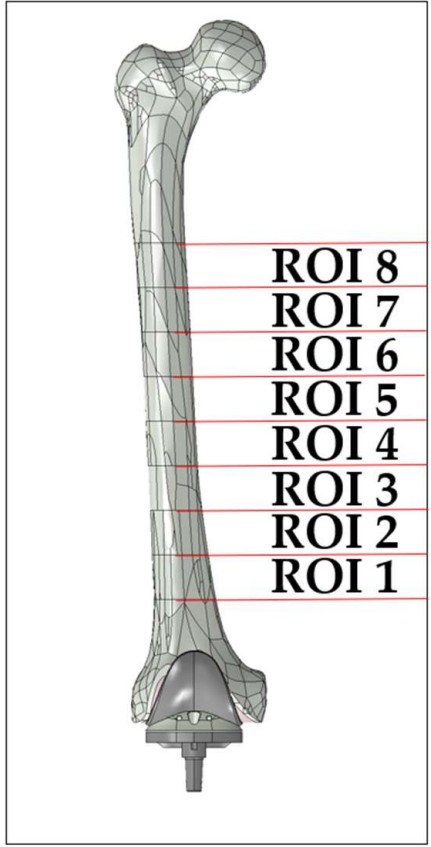

**Figure 2.** Regions of interest defined for the present study. All of the regions have a height of 30 mm measured along the femoral anatomical axis.

### 3. Results

Figure 3 concerns the cemented stem of 50 mm length, and reports the graphical overview of the average von Mises stress and contact pressure and area on the polyethylene insert for the different angles of deformity addressed.

From the figure, it is possible to note a change in the distribution of the stress and of the contact pressure on the tibial insert, that translates from lateral to medial when switching from valgus to varus configurations. Addressing the position of the contact point, the results show that the lateral contact is mainly located in the posterior section of the compartment while the medial contact point is mainly in the anterior one.

From a quantitative point of view, Tables 4 and 5 report, respectively, the values of medial and lateral contact forces and the values of the medial and lateral contact areas for the different configurations and for different stem lengths and cementing techniques.

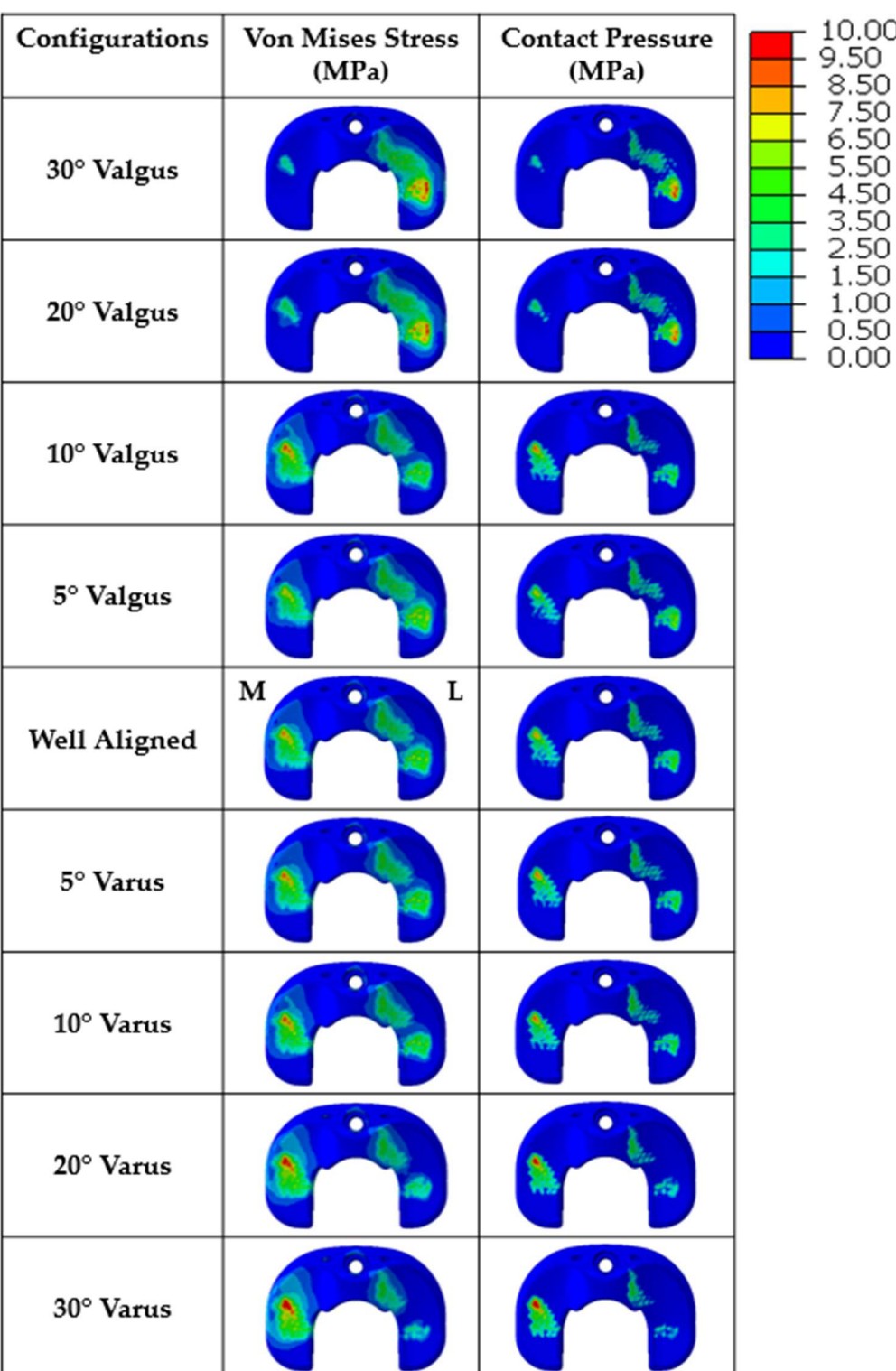

**Figure 3.** Qualitative overview of the von Mises stress distribution (in MPa) on the polyethylene insert and of the contact pressure (in MPa) and area on the polyethylene insert for the different varus/valgus configurations analyzed in the study, for a stem length of 50 mm cemented. M = medial, L = lateral.

**Table 4.** Medial and lateral contact forces for the different configurations and for different stem lengths and cementing techniques.

| Configuration | Press Fit 50 | | Cemented 50 | | Press Fit 120 | | Cemented 120 | |
|---|---|---|---|---|---|---|---|---|
| | Medial Force(N) | Lateral Force (N) | Medial Force(N) | Lateral Force(N) | Medial Force (N) | Lateral Force(N) | Medial Force (N) | Lateral Force (N) |
| 30° Valgus | 43 | 906 | 41 | 911 | 43 | 905 | 40 | 911 |
| 20° Valgus | 89 | 860 | 86 | 867 | 87 | 858 | 86 | 864 |
| 10° Valgus | 307 | 651 | 307 | 659 | 310 | 655 | 308 | 657 |
| 5° Valgus | 350 | 619 | 347 | 618 | 348 | 620 | 345 | 620 |
| Well Alligned | 442 | 532 | 447 | 532 | 443 | 528 | 445 | 532 |
| 5° Varus | 484 | 497 | 486 | 495 | 484 | 488 | 484 | 495 |
| 10° Varus | 517 | 469 | 516 | 462 | 515 | 456 | 517 | 464 |
| 20° Varus | 657 | 314 | 660 | 315 | 660 | 316 | 663 | 316 |
| 30° Varus | 726 | 243 | 726 | 244 | 728 | 243 | 731 | 246 |

**Table 5.** Medial and lateral contact areas for the different configurations and for the different stem lengths and cementing techniques.

| Configuration | Press Fit 50 | | Cemented 50 | | Press Fit 120 | | Cemented 120 | |
|---|---|---|---|---|---|---|---|---|
| | Medial Area (mm$^2$) | Lateral Area (mm$^2$) | Medial Area (mm$^2$) | Lateral Area (mm$^2$) | Medial Area (mm$^2$) | Lateral Area (mm$^2$) | Medial Area (mm$^2$) | Lateral Area (mm$^2$) |
| 30° Valgus | 15 | 157 | 15 | 158 | 15 | 159 | 15 | 158 |
| 20° Valgus | 28 | 156 | 28 | 156 | 28 | 155 | 28 | 156 |
| 10° Valgus | 62 | 132 | 63 | 132 | 62 | 132 | 63 | 132 |
| 5° Valgus | 71 | 130 | 71 | 130 | 71 | 130 | 70 | 130 |
| Well Alligned | 79 | 115 | 79 | 115 | 79 | 115 | 79 | 115 |
| 5° Varus | 81 | 110 | 81 | 110 | 81 | 109 | 81 | 109 |
| 10° Varus | 83 | 107 | 83 | 106 | 83 | 106 | 83 | 106 |
| 20° Varus | 95 | 91 | 96 | 91 | 95 | 91 | 96 | 91 |
| 30° Varus | 103 | 74 | 105 | 74 | 105 | 74 | 105 | 74 |

From these two tables, it is possible to note that the differences in terms of tibio-femoral contact relative to a change of stem lengths or fixation approach are negligible if compared to the changes induced by a different varus/valgus angle.

Figure 4 then reports, for the cemented 50 mm stem, different results relative to the medial side expressed as a percentage of their total on both sides; in detail, the different tibio-femoral contact forces and areas are represented in relation to the varus/valgus angles, together with the values of the input forces applied on the distal surface of the tibial insert (as a control for a comparative point of view).

It is possible to notice that each parameter presents a different trend, with the contact area being less sensible to changes of the varus/valgus angle (with a range of 9–58%) while the contact force is more sensible (with a range of 5–75%).

While the tibio-femoral contact forces and areas returned to be mostly insensible to variations of the stem length and fixation approach, the femoral bone showed to be influenced by these variations in terms of bone stress distributions in the different regions of interest addressed. Figure 5 reports the variations in terms of the average von Mises stress for ROI 1, 3, 5, and 7, according to the different configurations of stem length, fixation, and varus–valgus angle.

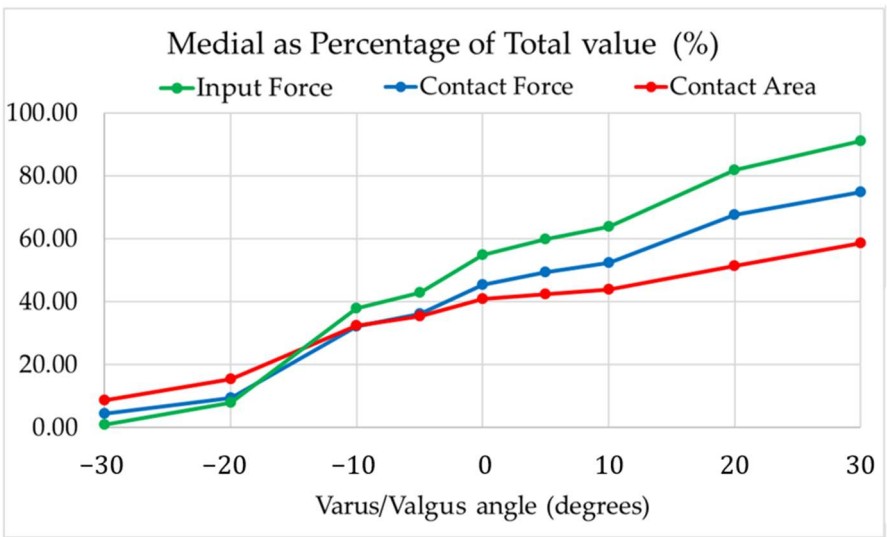

**Figure 4.** Input force, contact force and contact area relative to different varus/valgus angles for the cemented 50 mm stem.

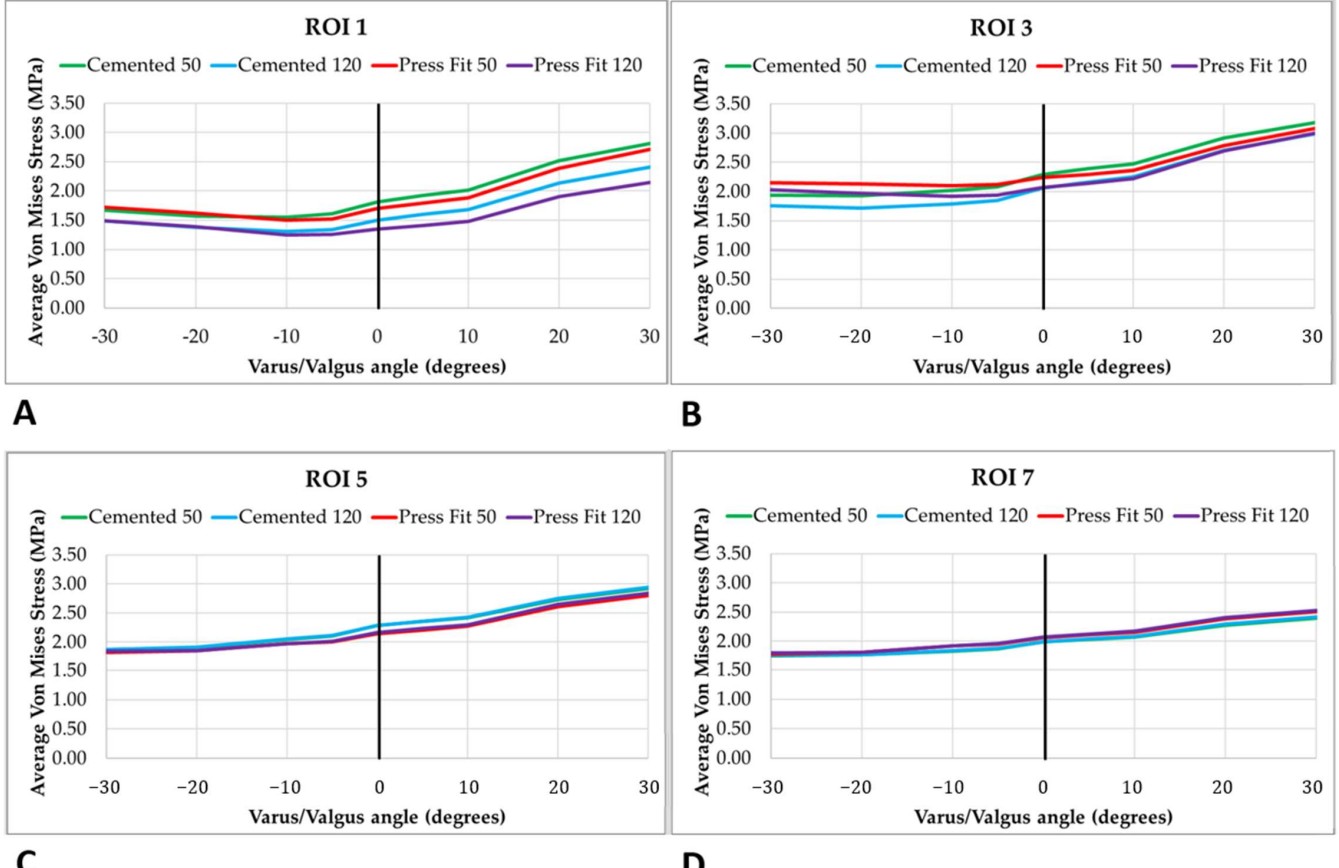

**Figure 5.** Average von Mises stresses in four regions of interest ((**A**): ROI 1, (**B**): ROI 3, (**C**): ROI 5, (**D**): ROI 7) for the different prosthesis features, according to the varus/valgus angles addressed.

The detailed values for the four different models are reported in Tables 6–9.

**Table 6.** Average von Mises stresses (in MPa) in the different ROIs for the 50 mm press-fit stem.

| Configuration | Press-Fit 50—Average von Mises Stress (MPa) | | | | | | | |
|---|---|---|---|---|---|---|---|---|
| | ROI 1 | ROI 2 | ROI 3 | ROI 4 | ROI 5 | ROI 6 | ROI 7 | ROI 8 |
| 30° Valgus | 1.73 | 1.88 | 2.15 | 1.82 | 1.83 | 1.72 | 1.79 | 1.83 |
| 20° Valgus | 1.62 | 1.80 | 2.14 | 1.81 | 1.86 | 1.73 | 1.81 | 1.84 |
| 10° Valgus | 1.50 | 1.73 | 2.10 | 1.81 | 1.97 | 1.75 | 1.93 | 1.87 |
| 5° Valgus | 1.52 | 1.77 | 2.12 | 1.85 | 2.01 | 1.78 | 1.96 | 1.87 |
| Well Aligned | 1.70 | 1.99 | 2.24 | 2.01 | 2.15 | 1.89 | 2.06 | 1.92 |
| 5° Varus | 1.79 | 2.09 | 2.29 | 2.10 | 2.21 | 1.95 | 2.11 | 1.94 |
| 10° Varus | 1.88 | 2.18 | 2.36 | 2.18 | 2.27 | 2.00 | 2.16 | 1.96 |
| 20° Varus | 2.39 | 2.69 | 2.79 | 2.63 | 2.61 | 2.28 | 2.39 | 2.08 |
| 30° Varus | 2.71 | 3.01 | 3.08 | 2.90 | 2.80 | 2.44 | 2.51 | 2.14 |

**Table 7.** Average von Mises stresses in the different ROIs for the 120 mm press-fit stem.

| Configuration | Press-Fit 120—Average von Mises Stress (MPa) | | | | | | | |
|---|---|---|---|---|---|---|---|---|
| | ROI 1 | ROI 2 | ROI 3 | ROI 4 | ROI 5 | ROI 6 | ROI 7 | ROI 8 |
| 30° Valgus | 1.49 | 1.65 | 2.03 | 1.84 | 1.85 | 1.74 | 1.81 | 1.81 |
| 20° Valgus | 1.39 | 1.57 | 1.97 | 1.82 | 1.86 | 1.72 | 1.81 | 1.81 |
| 10° Valgus | 1.25 | 1.46 | 1.92 | 1.82 | 1.98 | 1.75 | 1.93 | 1.83 |
| 5° Valgus | 1.26 | 1.50 | 1.94 | 1.86 | 2.02 | 1.78 | 1.96 | 1.85 |
| Well Aligned | 1.35 | 1.67 | 2.07 | 2.03 | 2.16 | 1.90 | 2.07 | 1.89 |
| 5° Varus | 1.41 | 1.77 | 2.14 | 2.12 | 2.23 | 1.96 | 2.13 | 1.92 |
| 10° Varus | 1.48 | 1.85 | 2.22 | 2.21 | 2.30 | 2.02 | 2.18 | 1.94 |
| 20° Varus | 1.90 | 2.32 | 2.70 | 2.68 | 2.65 | 2.31 | 2.40 | 2.06 |
| 30° Varus | 2.15 | 2.60 | 3.00 | 2.95 | 2.84 | 2.47 | 2.53 | 2.13 |

**Table 8.** Average von Mises stresses in the different ROIs for the 50 mm cemented stem.

| Configuration | Cemented 50—Average von Mises Stress (MPa) | | | | | | | |
|---|---|---|---|---|---|---|---|---|
| | ROI 1 | ROI 2 | ROI 3 | ROI 4 | ROI 5 | ROI 6 | ROI 7 | ROI 8 |
| 30° Valgus | 1.67 | 1.73 | 1.94 | 1.65 | 1.87 | 1.76 | 1.75 | 1.67 |
| 20° Valgus | 1.57 | 1.70 | 1.93 | 1.69 | 1.90 | 1.77 | 1.77 | 1.67 |
| 10° Valgus | 1.56 | 1.83 | 2.02 | 1.89 | 2.05 | 1.86 | 1.83 | 1.65 |
| 5° Valgus | 1.61 | 1.90 | 2.08 | 1.97 | 2.11 | 1.91 | 1.87 | 1.67 |
| Well Aligned | 1.81 | 2.14 | 2.29 | 2.21 | 2.29 | 2.06 | 1.99 | 1.72 |
| 5° Varus | 1.92 | 2.26 | 2.40 | 2.32 | 2.36 | 2.12 | 2.04 | 1.74 |
| 10° Varus | 2.02 | 2.36 | 2.48 | 2.40 | 2.42 | 2.17 | 2.08 | 1.76 |
| 20° Varus | 2.52 | 2.84 | 2.91 | 2.82 | 2.73 | 2.43 | 2.28 | 1.88 |
| 30° Varus | 2.81 | 3.13 | 3.18 | 3.06 | 2.92 | 2.58 | 2.40 | 1.95 |

**Table 9.** Average von Mises stresses in the different ROIs for the 120 mm cemented stem.

| Configuration | Cemented 120—Average von Mises Stress (MPa) | | | | | | | |
|---|---|---|---|---|---|---|---|---|
| | ROI 1 | ROI 2 | ROI 3 | ROI 4 | ROI 5 | ROI 6 | ROI 7 | ROI 8 |
| 30° Valgus | 1.49 | 1.53 | 1.75 | 1.66 | 1.88 | 1.76 | 1.76 | 1.67 |
| 20° Valgus | 1.38 | 1.48 | 1.72 | 1.69 | 1.91 | 1.78 | 1.77 | 1.67 |
| 10° Valgus | 1.30 | 1.56 | 1.79 | 1.89 | 2.05 | 1.87 | 1.84 | 1.65 |
| 5° Valgus | 1.34 | 1.62 | 1.85 | 1.97 | 2.11 | 1.91 | 1.88 | 1.67 |
| Well Aligned | 1.50 | 1.84 | 2.06 | 2.21 | 2.29 | 2.05 | 1.99 | 1.72 |
| 5° Varus | 1.60 | 1.95 | 2.16 | 2.32 | 2.36 | 2.12 | 2.04 | 1.74 |
| 10° Varus | 1.68 | 2.04 | 2.25 | 2.41 | 2.43 | 2.18 | 2.09 | 1.77 |
| 20° Varus | 2.14 | 2.50 | 2.70 | 2.83 | 2.75 | 2.44 | 2.30 | 1.89 |
| 30° Varus | 2.40 | 2.78 | 2.99 | 3.08 | 2.94 | 2.60 | 2.42 | 1.96 |

From Figure 5, it is thus possible to see that each model induces a different stress distribution, especially in the distal femoral region (ROI 1), and that the differences are

instead lower in the proximal regions (ROI 7). Moreover, it is also possible to see that, for each configuration analyzed, the change in average von Mises stress is mostly insensible to changes of the valgus angle (with a variation curve almost horizontal) while it is moderately sensible to changes in terms of the varus angle.

Analyzing the proximal regions in detail, it is possible to see that in ROI 1 the short stem is characterized by a higher average stress (almost the same for cemented and press-fit fixation) while the press-fit long stem is characterized by lower stresses, especially in the varus conditions: this outcome is reasonably due to the stress shielding effect related to the longer stems, which consequently leads to slightly higher stresses in the regions close to the stem tip.

## 4. Discussion

In this study, a series of finite element simulations was performed in order to evaluate the effects on the insert and bones of different configurations of RH TKA prosthesis, in terms of valgus and varus deformities, lengths of femoral stems, and fixation approaches.

The results found returned that, overall, the tibio-femoral contact forces and areas are not considerably influenced by variations of stem length and fixation approach while varus/valgus variations lead to relative changes (varying according to the parameter analyzed and to the gravity of the deformity). Moreover, the femoral bone stress distribution was determined to be influenced by the variations in the prosthesis design and fixation, with further differences found in relation to the region of interest considered.

Addressing more in depth, the outcomes of the tibio-femoral interface, the reason behind the lower sensibility of the contact area to varus/valgus angles and the overall insensibility to the variations in stem length and fixation approach is indeed to be found in the design of the RH TKA itself. In detail the central constraint, which allows internal-external rotations and superior-inferior translations, does not enable any other degree of freedom and thus contributes to maintain the tibio-femoral interactions to be more constant. For this reason, indeed, changes in the contact area are remarkably low in varus deformities ranging from 0° to 30° (less than 20% variation in terms of contact area, while slightly higher values are found in terms of force), even if the difference in the input forces is greater than 35%. These outcomes are then in agreement with several clinical studies focused on endo-model hinged TKAs, which reported its stability and absence of cases of wear due to overloading of the polyethylene [13,15,49]. Moreover, a recently published study [50] on the use of the rotating hinge and followed over 10-years concluded that using a specific RH TKA design with less rotational constraint has better clinical and survival outcomes than implants with greater rotational constraint (such as one specific CCK addressed in the study) in case of varus and valgus deformities.

Addressing then the outcomes in terms of femoral bone stress, the results highlighted how these values are influenced more by the stem length and fixation rather than the values of varus and valgus deformities; these results are indeed in agreement with that found in the literature concerning the effects of stem design [20,21] and moreover demonstrate the ability of this prosthesis to maintain similar outcomes in terms of bone-prosthesis interactions, despite the different levels of deformity of the patient joint.

It is however to be mentioned that this study presents a series of limitations. Firstly, only one implant model was considered, with a single size and typology being analyzed; a single angle of flection was then simulated for all of the configurations. These two restrictions may indeed represent a limitation for the generalization of the results, but it is to be highlighted that these choices allowed to perform a comparative study focused on the specific parameters taken into consideration, thus obtaining meaningful information that can therefore represent an interesting insight for the surgeons during the decision-making process. Addressing the finite element models, it is to be mentioned that the material models used to simulate the bones in this study implied a series of assumptions: indeed, no bone activity was considered in the simulations and therefore the variations in femoral mechanic characteristic, usually occurring in response to the different loading conditions

the joint undergoes [51–54], are not taken into account. Despite these latter limitations, however, these models were developed based on previously published and validated ones [21] and therefore the results they provided can be considered reliable, as they are furthermore in agreement with the literature [2,20,21].

Concerning the validation of the model, it is to be mentioned that no direct validation was performed for this study; however, it is to be considered that the model used was implemented starting from validated models that can be found in previously published studies [21,25–27], and therefore can be considered as reliable in its results.

## 5. Conclusions

This study provided interesting information on the influence of varus/valgus deformities of different levels of severity on the performances of different configurations of rotating hinged TKA, showing how the prosthesis design features mainly alter the bone stress distribution while varus/valgus deformities are the main responsible factors for variations in tibio-femoral contact forces and areas.

This result represents an interesting information for the surgeons: for implants characterized by high level of constraint, indeed, alterations in the bending and torsional stiffness and moments might occur if the stem length and fixation are modified, and they would therefore be transferred to the tibio-femoral interface of the implant; this eventuality may thus condition the surgeon's choice, which may aim for a compromise in order to avoid any tibio-femoral issues deriving from these alterations. The results of this study showed instead that, in the case of the rotating hinge analyzed, no remarkable mechanical consequences on the tibio-femoral interface are found despite variations in stem and fixation configuration; the surgeon can therefore focus their decisions on optimizing the bone-implant interface, without the need for finding a compromise in the fear of altering excessively the tibio-femoral biomechanics.

**Supplementary Materials:** The following supporting information can be downloaded at: https://www.mdpi.com/article/10.3390/app13042738/s1, Convergence Test.

**Author Contributions:** Conceptualization, E.B. and B.I.; methodology, B.I.; software, B.I.; formal analysis, E.B. and B.I.; investigation, E.B. and B.I.; data curation, B.I.; writing—original draft preparation, E.B. and B.I.; writing—review and editing, E.B.; visualization, B.I.; supervision, B.I.; project administration, B.I. All authors have read and agreed to the published version of the manuscript.

**Funding:** This research received no external funding.

**Institutional Review Board Statement:** Not applicable.

**Informed Consent Statement:** Not applicable.

**Data Availability Statement:** Not applicable.

**Conflicts of Interest:** The authors declare no conflict of interest.

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
