# Peer review of "Biomechanical Analysis of Femoral Stem Features in Hinged Revision TKA with Valgus or Varus Deformity: A Comparative Finite Elements Study"

_applsci, doi:10.3390/app13042738_

Round 1
Reviewer 1 Report
The present study intends to help and support clinicians and specialists in the decision process concerning the length and type of stems to use in case Varus-Valgus deformity is present in a patient.
There might be a clearer description in the Conclusion chapter regarding that, does this help clinicians and specialists?
There are to many digits in Tables 4 and 5.
Is the Poission ratio 0.58 >0.5 accurate?
The boundary conditions are not specified adequately.
Author Response
The authors thank the reviewer for the time given to revise and to provide their feedback for the manuscript.
The modifications done in the manuscript are highlighted in yellow, to help trackability.
In the following response, the comments of the reviewer will be reported in bold, and the extracts from the revised manuscript will be reported in italics.
The present study intends to help and support clinicians and specialists in the decision process concerning the length and type of stems to use in case Varus-Valgus deformity is present in a patient.
There might be a clearer description in the Conclusion chapter regarding that, does this help clinicians and specialists?
The authors agree with the reviewer’s comment, the conclusion section was modified and improved accordingly, adding the following paragraph:
This result represents an interesting information for the surgeons: for implants characterized by high level of constraint, indeed, alterations in the bending and tor-sional stiffness and moments might occur if the stem length and fixation are modified, and they would therefore be transferred to the tibio-femoral interface of the implant; this eventuality may thus condition the surgeon’s choice, which may aim for a com-promise in order to avoid any tibio-femoral issues deriving from these alterations. The results of this study showed instead that, in the case of the Rotating hinge analyzed, no remarkable mechanical consequences on the tibio-femoral interface are found despite variations in stem and fixation configuration; the surgeon can therefore focus their de-cisions on optimizing the bone-implant interface, without the need of finding a com-promise in the fear of altering excessively the tibio-femoral biomechanics.
There are to many digits in Tables 4 and 5.
The authors agree with the reviewer, the tables were modified accordingly.
Is the Poission ratio 0.58 >0.5 accurate?
The authors apologize for the typo, indeed the value considered was 0.51 and not 0.58 as reported. For what concerns the fact that the value used is higher than 0.5, the authors want to highlight that this phenomenon may occur in extreme cases of non-homogeneous and anisotropic materials, due to the specificity of their structure; this is indeed what happens in case of the bone as it is considered for this model (i.e. transversally isotropic), and therefore this value was extracted from the literature and used.
The boundary conditions are not specified adequately.
The paragraph was modified accordingly, and more detail was provided to enhance manuscript clarity.
Here are reported the sentences added in the relative section:
Each configuration analyzed was tested under the same total load conditions, i.e. the proximal part of the femur constrained and a total vertical compressive force of 1,000 N applied to the inferior face of the tibial tray. The subdivision of this total force on the two sides of the tibial tray was then used as a parameter in order to model the different levels of Varus/Valgum deformities: a simplified model in the literature [11] allowed indeed to obtain the percentage of force distribution in the medial and lateral regions of the tibial component for each angle of deformity (based on the equilibrium of forces and moments in the frontal plane), and was therefore taken as reference. The percentages of force distribution in medial and lateral compartment obtained from this mathematical model are reported in Table 3.
Reviewer 2 Report
The manuscript presents a pertinent finite element study concerning the effect of varus/valgus load configurations in the bones when a hinged total knee arthroplasty (TKA) is used. The work highlighted the important role of parametric numerical models for supporting the decision process of clinicians in case Varus-Valgus deformity affects a patient. The study is relevant to be published in the journal.
The state of the art is well-presented. The overall presentation of the FEM and FEM-results is very clear. The Authors also identify the limitations of their model.
Specific comments:
Line 305: I recommend substituting “info” with "information" for a more rigorous language
Author Response
The authors thank the reviewer for the time given to revise and to provide their feedback for the manuscript.
The modifications done in the manuscript are highlighted in yellow, to help trackability.
In the following response, the comments of the reviewer will be reported in bold, and the extracts from the revised manuscript will be reported in italics.
The manuscript presents a pertinent finite element study concerning the effect of varus/valgus load configurations in the bones when a hinged total knee arthroplasty (TKA) is used. The work highlighted the important role of parametric numerical models for supporting the decision process of clinicians in case Varus-Valgus deformity affects a patient. The study is relevant to be published in the journal.
The state of the art is well-presented. The overall presentation of the FEM and FEM-results is very clear. The Authors also identify the limitations of their model.
The authors deeply thank the reviewer for their kind words.
Specific comments:
Line 305: I recommend substituting “info” with "information" for a more rigorous language
The authors agree with the comment, the manuscript was modified accordingly.
Reviewer 3 Report
One of the main objectives of the study is to investigate the effect of stem length and cement on tibio-femoral contact biomechanics. These study shows that these parameters have no impact. It is unclear why the authors look at these parameters in the first place, as it seems evident that they will not have any effect. Cement and stem length are far away from the insert and it does not matter how the stem is held in the bone (cement vs uncemented, or stem length). I can see how varus and valgus may influence the bone stress depending on stem length or cement, however, this is not what was done in this study. The manuscript would benefit from being more focused. For example, the paper could focus on the effect on varus-valgus only, on contact force, contact area and bone stress. Potentially, bone stress could be investigated for the two stem lengths and cement, but joint contact parameters should not.
The study provides a lot of results, but no guidelines for the surgeon. Should varus-valgus malpositioning be avoided? What are acceptable ranges? Is there an optimum?
The language must be improved. Many sentences are difficult to understand, and many are not well written (a few examples on lines 36-38, 49-51, 112-114, 179-181, etc). The whole manuscript must be edited by an native english speaker.
There are 16 (!) self-citations, amounting to a third of the references. That is clearly a bit too much. There are certainly papers from other authors that are also relevant?
Author Response
The authors thank the reviewer for the time given to revise and to provide their feedback for the manuscript.
The modifications done in the manuscript are highlighted in yellow, to help trackability.
In the following response, the comments of the reviewer will be reported in bold, and the extracts from the revised manuscript will be reported in italics.
One of the main objectives of the study is to investigate the effect of stem length and cement on tibio-femoral contact biomechanics. These study shows that these parameters have no impact. It is unclear why the authors look at these parameters in the first place, as it seems evident that they will not have any effect. Cement and stem length are far away from the insert and it does not matter how the stem is held in the bone (cement vs uncemented, or stem length). I can see how varus and valgus may influence the bone stress depending on stem length or cement, however, this is not what was done in this study. The manuscript would benefit from being more focused. For example, the paper could focus on the effect on varus-valgus only, on contact force, contact area and bone stress. Potentially, bone stress could be investigated for the two stem lengths and cement, but joint contact parameters should not.
The authors thank the reviewer for their comment, and want to highlight that the main rationale of the study is to evaluate the results of the use of different rotating-hinge configurations in cases of joint deformities, addressing both tibio-femoral and implant bone interfaces, as both are important for the overall success of the implant. The fact that the results showed that changes in the stem configuration were altering only marginally the tibio-femoral interface is not completely evident from the start, mainly when dealing with a constrained prosthesis: changes in the bending and torsional stiffness and moments can indeed occur if the stem length or fixation are modified, and these moments would consequently be transferred to the tibio-femoral interface of the implant modifying this interface as well. The results of this study lead therefore to a positive conclusion, as the surgeon is allowed to focus on optimizing the bone-implant interface without having to keep in consideration eventual remarkable mechanical consequences on the tibio-femoral interface. The rationale and conclusions of the study were revised in order to improve their clarity, and therefore justify the choices made by the authors in the light of the comments made by the reviewer.
Rationale section:
Despite its advantages, this kind of implant also implies the relative risks of mechanical failure and infection [9,22–24], which can furthermore increase in case of eventual patient joint deformities. The level of constraint provided by the rotating hinge, paired with the use of different stems and/or fixation approached, may therefore lead to eventual over-constraining issues affecting the tibio-femoral and/or the bone-implant interactions; a more comprehensive guideline on the suggested use and optimal configuration of these implants (when mainly aimed to address high deformities) is however still missing.The present study intends to help and support clinicians and specialists, assisting the decision process concerning the length and type of stems to use in case Varus-Valgus deformity is present in a patient.
Conclusion Section:
This result represents an interesting information for the surgeons: for implants characterized by high level of constraint, indeed, alterations in the bending and torsional stiffness and moments might occur if the stem length and fixation are modified, and they would therefore be transferred to the tibio-femoral interface of the implant; this eventuality may thus condition the surgeon’s choice, which may aim for a compromise in order to avoid any tibio-femoral issues deriving from these alterations. The results of this study showed instead that, in the case of the Rotating hinge analyzed, no remarkable mechanical consequences on the tibio-femoral interface are found despite variations in stem and fixation configuration; the surgeon can therefore focus their decisions on optimizing the bone-implant interface, without the need of finding a compromise in the fear of altering excessively the tibio-femoral biomechanics.
The study provides a lot of results, but no guidelines for the surgeon. Should varus-valgus malpositioning be avoided? What are acceptable ranges? Is there an optimum?
Accordingly to what mentioned in the previous point, one of the most important conclusions that can be made from this study results consists in the fact that the surgeon can focus their decisions in terms of stem length and fixation in order to optimize the bone-implant interface, with the knowledge that no remarkable biomechanical consequence will occur on the tibio-femoral interface because of these choices. The relative sentence was added in the conclusion section to improve the clarity of the manuscript.
The language must be improved. Many sentences are difficult to understand, and many are not well written (a few examples on lines 36-38, 49-51, 112-114, 179-181, etc). The whole manuscript must be edited by an native english speaker.
The authors thank the reviewer for their comment, the readability of the manuscript was improved through an extensive revision of this latter.
There are 16 (!) self-citations, amounting to a third of the references. That is clearly a bit too much. There are certainly papers from other authors that are also relevant?
The authors want to highlight that our group has a relevant long-time experience in the field, in detail for what concerns revision TKA implants. For this reason, many of the references mentioned by the reviewer are relative to Material and Methods from studies previously performed by the group itself (on which the present article is also based), that were therefore considered interesting for the reader to better understand the decisions and modelling approaches involved in the presented activity. Nonetheless, different references were added to the manuscript (increasing from 45 to 54) during the revision to improve its overall quality.
Reviewer 4 Report
The authors present interesting finite element analyses to determine the effect of varus/valgus load configurations in bones when a hinged TKA is adopted.
The work could be of potential interests both in the biomechanical and in the clinical field.
However, some aspects are still lacking and are required to improve the quality of the manuscript:
1) Introduction: the authors lack in mentioning the complexity of bone structure, that is able to actively respond to a wide range of loading scenarios. Indeed, several recent works are addressing this topic (https://doi.org/10.1016/j.matdes.2022.111486, https://doi.org/10.3390/ma14051240, https://doi.org/10.1016/S0021-9290(01)00069-0) and it is worth discussing this point to justify the need of numerical modelling strategies.
Additionally, the authors did not mention the fact that femoral mechanical characteristics highly vary in response to the external loading condition, as described in detail in http://dx.doi.org/10.2139/ssrn.3878345 . For sure, the numerical modeling strategies the authors adopted would require some simplifications, however, the assumptions the authors made in the material and methods section should be justified by introducing a short paragraph in the introduction that includes the complexity of femoral architecture and the need for simplification in numerical modelling.
2) Materials and methods: did the authors perform a mesh convergence analysis? Please include it in the manuscript, to verify the reliability of the obtained observations.
3) Results: the authors present the results in terms of von Mises stresses, that are basically average stresses. What is the outcome in terms of principal stress components? This is a core point to be considered, for determining critical regions.
Author Response
The authors thank the reviewer for the time given to revise and to provide their feedback for the manuscript.
The modifications done in the manuscript are highlighted in yellow, to help trackability.
In the following response, the comments of the reviewer will be reported in bold, and the extracts from the revised manuscript will be reported in italics.
The authors present interesting finite element analyses to determine the effect of varus/valgus load configurations in bones when a hinged TKA is adopted.
The work could be of potential interests both in the biomechanical and in the clinical field.
However, some aspects are still lacking and are required to improve the quality of the manuscript:
1) Introduction: the authors lack in mentioning the complexity of bone structure, that is able to actively respond to a wide range of loading scenarios. Indeed, several recent works are addressing this topic (https://doi.org/10.1016/j.matdes.2022.111486, https://doi.org/10.3390/ma14051240, https://doi.org/10.1016/S0021-9290(01)00069-0) and it is worth discussing this point to justify the need of numerical modelling strategies. Additionally, the authors did not mention the fact that femoral mechanical characteristics highly vary in response to the external loading condition, as described in detail in http://dx.doi.org/10.2139/ssrn.3878345 . For sure, the numerical modeling strategies the authors adopted would require some simplifications, however, the assumptions the authors made in the material and methods section should be justified by introducing a short paragraph in the introduction that includes the complexity of femoral architecture and the need for simplification in numerical modelling.
The authors thank the reviewer for their comment regarding the representation of the bone mechanical behavior and its crucial role in the simulations; modifications to the manuscript were made by adding a relative sentence in the Discussions (in detail, in the limitation section), in order to address more clearly the simplifications made in the modeling. This choice was deemed the most suitable one, as the assumptions made in the modelling of the materials primarily represent a limitation of the study, rather than part of the introduction itself. Additionally, the references suggested by the reviewer were taken into consideration and added in the same section, alongside the references used to justify the simplifications made in the study.
Addressing the finite element models, it is to be mentioned that the material models used to simulate the bones in this study implied a series of assumptions: indeed, no bone activity was considered in the simulations and therefore the variations in femoral mechanic characteristic, usually occurring in response to the different loading condi-tions the joint undergoes [46–49], are not taken into account; furthermore, no direct validation was performed in this study. Despite these latter limitations, however, these models were developed based on previously published and validated ones [21] and therefore the results they provided can be considered reliable, as they are furthermore in agreement with the literature [2,20,21].
2) Materials and methods: did the authors perform a mesh convergence analysis? Please include it in the manuscript, to verify the reliability of the obtained observations.
The authors thank the reviewer for the comment, the convergence analysis addressing the bone-implant interaction was added to the paper as supplementary material in order to improve the completeness of the manuscript. For what concerns the convergence analyses performed for the insert and the internal components of the implant, these operations were already performed in the previous studies referenced in the relative paragraph [2, 20] and therefore the relative analysis was not performed again.
A refinement of mesh was performed in the contact area of internal components of the implant, and in the tibio-femoral and bone-implant interfaces sections to make sure that the selected mesh was the proper one to achieve the sought results: to check the quality of the mesh and the proper size, a convergence test was thus performed [2,20] and is available in the supplementary material.
3) Results: the authors present the results in terms of von Mises stresses, that are basically average stresses. What is the outcome in terms of principal stress components? This is a core point to be considered, for determining critical regions.
The choice to use the von Mises stress to evaluate and compare the results of the simulations performed was made for several reasons: von Mises stress is a well-established and widely used measure for evaluating the overall stress state in bones and prostheses, as reported in many studies in the literature [44-48]; it furthermore provides a single scalar value, therefore ideal to easily compare the results of different simulations as required by the present study. In conclusion, the choice to use the von Mises stress was based on its established use as a measure of stress, its ease of comparison, and its ability to indicate eventual failure in materials.